# Locality Sensitive Teaching

**Zhaozhuo Xu**
Rice University
zx22@rice.edu

**Beidi Chen**
Stanford University
beidic@stanford.edu

**Chaojian Li**
Rice University
chaojian.li@rice.edu

**Weiyang Liu**
University of Cambridge and MPI-IS Tübingen
wl396@cam.ac.uk

**Le Song**
Biomap and MBZUAI
dasongle@gmail.com

**Yingyan Lin**
Rice University
yingyan.lin@rice.edu

**Anshumali Shrivastava**
Rice University and ThirdAI Corp.
anshumali@rice.edu

## Abstract

The emergence of the Internet-of-Things (IoT) sheds light on applying the machine teaching (MT) algorithms for online personalized education on home devices. This direction becomes more promising during the COVID-19 pandemic when in-person education becomes infeasible. However, as one of the most influential and practical MT paradigms, iterative machine teaching (IMT) is prohibited on IoT devices due to its inefficient and unscalable algorithms. IMT is a paradigm where a teacher feeds examples iteratively and intelligently based on the learner's status. In each iteration, current IMT algorithms greedily traverse the whole training set to find an example for the learner, which is computationally expensive in practice. We propose a novel teaching framework, Locality Sensitive Teaching (LST), based on locality sensitive sampling, to overcome these challenges. LST has provable near-constant time complexity, which is exponentially better than the existing baseline. With at most $425.12\times$ speedups and $99.76\%$ energy savings over IMT, LST is the first algorithm that enables energy and time efficient machine teaching on IoT devices. Owing to LST's substantial efficiency and scalability, it is readily applicable in real-world education scenarios.

## 1 Introduction

During the COVID-19 pandemic, there is an increasing demand for learning at home. Computer-based personalized education (CBPE) [1, 2, 3] on Internet-of-Things (IoT) devices become essential as it improves the accessibility of students to the educational resources and reduces the potential privacy risks. Due to the popularity of Coursera, Duolingo, and EdX, online on-device education has become increasingly more important nowadays. To better understand and improve the CBPE approaches in this context, we consider machine teaching (MT) [4] as a simplified yet helpful paradigm. Specifically, MT defines the problem where a machine teacher constructs a minimal set of examples that allows a student to learn a target concept repeatedly. However, MT on the IoT device is still prohibitive for two primary reasons: (1) MT technique does not allow real-time interaction between teachers and students. It is designed to work over a static set and cannot incorporate real-time student feedback, (2) MT can not support on-device training due to the expensive cost spent in constructing this minimal teaching set.

35th Conference on Neural Information Processing Systems (NeurIPS 2021).

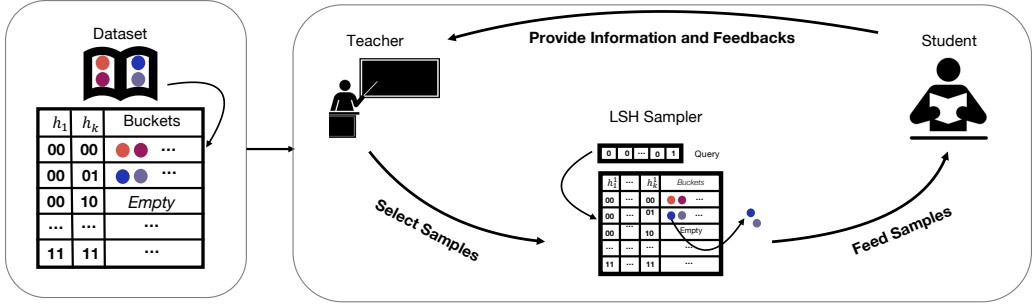

1. Data Preprocessing        2. Iterative Teaching

Figure 1: A brief overview of the proposed LST algorithm. There are two stages: 1) during data preprocessing, the indices of data samples are stored in LSH hash tables. 2) during the teaching stage, in each iteration, the machine teacher uses the student's information to query an example from hash tables. Then the machine teacher feeds the example to the student.

As a result, a more practical paradigm – iterative machine teaching (IMT) [5, 6] has been proposed to achieve state-of-the-art teaching performance. In IMT, the teacher interacts with the student in every iteration and aims to teach the target concept with a minimum number of iterations. Although IMT reduces both teaching set size and teaching iterations, it is still not efficient enough. The time and energy required by IMT are prohibitively expensive on IoT devices. There are two primary reasons: (1) IMT is not designed for real-time teaching. Current IMT uses a deterministic greedy algorithm, which traverses the entire dataset to find the optimal teaching example during each iteration. This linear time scan costs excessive time in real-world datasets, preventing more frequent interactions between machine teachers and students. (2) IMT has poor energy efficiency because the greedy scan requires scoring and ranking on the training examples. The energy budget of IoT devices such as integrated CPU-GPU System-on-Chips (SoCs) cannot afford this expensive energy consumption in real-world teaching. Clearly, there is a gap between IMT and its deployment on IoT devices.

Contemporary IMT literature focuses on developing and analyzing theories, models, or paradigms in teaching. Few studies are addressing the trade-offs between teachability and efficiency. Although some IMT algorithms manage their potential applications in crowd-sourcing [7, 8, 9], personal education [4], or online data poison [10, 11], they do not consider the edge computing and IoT settings and ignore the substantial potential impact behind it.

Our work proposes a time and energy-efficient IMT algorithm, namely Locality Sensitive Teaching (LST), that enables machine teaching on IoT devices. A real-life teaching observation inspires LST: a human teacher teaches a target concept to the student efficiently based on his or her knowledge in mind rather than a brutal-force scan over all the course materials. Similarly, an efficient IMT algorithm should provide optimal examples by looking up from a data structure rather than traversing the entire example set. Our LST algorithm effectively pre-indexes the teaching examples in hash tables. Then, given the student's current state at each iteration, we regard it as a query and generate examples through efficient sampling using the hash tables [12]. Moreover, we significantly improve the time and energy efficiency by reducing the expensive linear scan into a lightweight lookup in hash tables. As a result, LST could be compatible with edge computing on IoT devices.

Our proposal, LST, comes with three key questions: (1) how to reformulate the IMT problem as a sampling problem that involves efficient hash table data structures? (2) how to maintain the identical teachability of IMT when utilizing the efficient hash table structures? (3) how to provide an efficient LST implementation on IoT devices that reduce energy and time consumption?

At a high level, we tackle these challenges as follows:

- We reformulate IMT as an adaptive inner product sampling problem. We partition the teaching materials into two parts: (1) query: current state of student and the optimal model possessed by the machine teacher, (2) data: teaching examples denoted as feature-label pairs. We apply an asymmetric transformation that projects both parts as vectors. Then, we argue that the original IMT formula can be formulated as an adaptive inner product sampling. Given the query vector, the task is to sample data vectors with large inner products. Therefore, locality sensitive sampling can be introduced for efficient teaching examples generation.

- We demonstrate, both theoretically and empirically, that LST preserves the teachability of IMT. Theoretically, we prove the LST can achieve exponential teachability with high probability. Empirically, the experiments on real-world teaching indicate that LST matches or even exceed the teachability of original IMT.
- We provide a novel LST system design on integrated CPU-GPU SoC platforms that performs time and energy-efficient teaching in real-world settings. We re-partition the locality sensitive sampling procedure into two parts: GPU-friendly random projection and CPU-friendly hash table lookups. Then, we exploit the benefits of fast matrix multiplications on GPU and efficient hash table lookups implemented on CPUs. Therefore, our LST system takes full advantage of the memory-constrained CPU-GPU SoCs.

Through extensive experiments in real-world teaching scenarios, we demonstrate that LST performs exponential teachability that matches or even exceeds IMT while achieving at most $425.12\times$ speedups and $99.76\%$ energy savings on IoT devices. On server-based evaluation, LST achieves more than $2000\times$ speedups over IMT.

## 2 Related Work and Preliminaries

### 2.1 Iterative Machine Teaching

In machine teaching, a machine teacher obtains a minimal training set for a student to learn a concept. [4, 13] introduce a general teaching framework and build its connections to curriculum learning [14]. [15] studies the machine teaching for Bayesian learners in exponential family and provides the teaching example by solving an optimization problem. [16] gives the linear learners' teaching dimension. Machine teaching has been found useful in cybersecurity [17], human-computer interaction [18], and human education [19]. [20, 6, 21] study the teaching scenario where the teacher aims to teach multiple concepts to a forgetful learner. [22, 6] assume that the learner is black-box and study how the teaching should be performed. [23, 6] discuss how the teacher can teach multiple different learners. [24, 25, 26] connect the machine teaching problem with inverse reinforcement learning by studying an inverse reinforcement learner. [27, 2] consider the machine teaching from an interpretable perspective. [28, 29] discuss the teaching model with a version space learner. [3] provides machine teaching extensions in human-in-the-loop settings. [30, 31, 32] study the machine teaching in theory.

Previous machine teaching works focus on the improvement of teachability. [5] proposes the iterative teaching paradigm. In this paradigm, an omniscient teacher model knows almost everything about the student. Then, the omniscient teacher provides training examples based on the student's status. [33] further generalizes this teaching scenario and introduces an optimal control approach to address sequential machine teaching. Built upon the same framework as [5], [6] looks into the IMT problem when the learner becomes black-box. [34] proposes to perform iterative teaching by label synthesis while still preserving the provable teaching speedup. In our work, we address the scalability bottleneck of IMT by leveraging the power of locality sensitive hashing.

This paper follows the same teaching setting as [5]. Without loss of generality, we mainly discuss the omniscient teaching model where the teacher knows the learner's full information. We apply our LST to speed up the omniscient teacher. We note that it is straightforward to improve the teaching quality and efficiency of more advanced teaching models using our proposed method.

### 2.2 Hashing-based Sampling

This section briefly describes the recent development of efficient sampling and estimation via locality sensitive hashing (LSH) [35, 36, 37, 38, 39, 40, 41, 42, 43, 44, 45, 46]. The intuition of LSH is to hash similar items into the a bucket of a hash table via functions such as random projection. For details of LSH, we refer to Appendix A.

We denote $p$ as the collision probability of LSH function. Therefore, given a query, an item will be retrieved with probability $p$. The precise form of $p$ is determined by the LSH function family. This sampling view of LSH has been used in a wide range of applications, such as neural networks training [47, 48], kernel density estimation [49], outlier detection [50], and optimization [51].

The success of LSH in efficient inner product search sheds light on its application in adaptive sampling. More specifically, in this paper, given a collection $\mathcal{C}$ of input and a query $Q$, candidates $S$ are

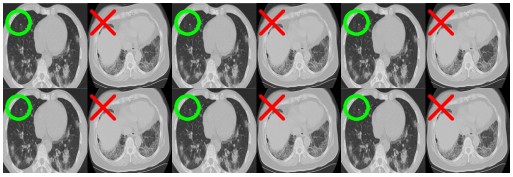
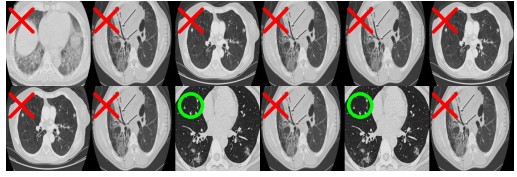

| (a) Vanilla IMT | (b) Our Proposed LST |

Figure 2: A motivating example on COVID-19 Teaching via CT images. The first 12 examples selected by IMT and LST examples are presented on (a) and (b). We mark each COVID image with a green circle and each regular image by the red cross. We observe that our proposed LST selects diverse, confusing images while IMT keeps selecting the same two images. Details are shown in Appendix D.

drawn from $\mathcal{C}$, in which each $x_i \in \mathcal{C}$ is sampled with $p_i \leq 1$, using LSH algorithm with special designed hashing [52]. Note here $p_i$ is a monotonically increasing function of $Q \cdot x_i$. Therefore, once we pay a linear preprocessing cost that maps $\mathcal{C}$ into hash tables, the adaptive sampling for query $Q$ could be done via a few hash lookups. The LSH sampler component in Figure 1 demonstrates the sampling process using LSH. Similar to what has been mentioned in [51], the sampling scheme is not a valid distribution, i.e., $\sum_{x_i \in \mathcal{C}} p_i \neq 1$. Given a query, the probability of sampling $x_i$ is not independent of the probability of sampling $x_j$ ($i \neq j$). However, we can still use it for unbiased estimation. $p_i$ is a monotonic function of $Q \cdot x_i$ because $p_i = (1 - (1 - g(Q \cdot x_i))^K)^L$, where $g(Q \cdot x_i)$ is the collision probability.

## 3 Locality Sensitive Teaching

This section first shows that finding a teaching example in the current IMT can be reformulated as an adaptive inner product sampling (AIPS) problem. Then we propose our LST algorithm to effectively and efficiently perform this adaptive sampling via Locality Sensitive Sampling (LSS).

### 3.1 A Motivating Example

To showcase the significance of LST, we conduct an experiment on teaching students to recognize COVID-19 from CT images. We apply both LST and IMT on a binary-class CT image dataset [53] (labeled as normal or COVID-19, details are in Appendix D). After these 12 examples, LST achieves less loss than IMT. Figure 2 shows the comparison of teaching examples. One can observe that LST's teaching examples yield more diversity because of the randomness it introduces. In contrast, IMT simply repeats two examples without giving a broader view of the other teaching set examples. This phenomenon suggests LST's potential to achieve more intuitively plausible teaching.

### 3.2 Reformulating IMT as Adaptive Sampling

Following a similar notation from [5], we denote $N$ as the number of samples, and $S = \{(x_i, y_i) | 0 < i \leq N\}$ as the input samples along with their labels . Then, we construct a student model with loss function $\ell$ and learning rate $\eta$. Given the model, we denote $w_t$ as the parameter of the student model at iteration $t$. $w^*$ is the optimal parameter obtained by the omniscient teacher. In each iteration, IMT selects an example by performing the following greedy algorithm:

**Problem 1** (**Greedy Teaching Algorithm (GTA)** ). *Given a large set $S$, the optimal parameter $w^*$ and model parameter $w_t$, we want to solve the following optimization problem:*

$$x^t, y^t = \arg \min_{(x,y) \in S} \eta^2 \|\nabla l\|_2^2 - 2\eta \langle w^t - w^*, \nabla l \rangle, \tag{1}$$

*where we have $\nabla l = \frac{\partial \ell(\langle w^t, x \rangle, y)}{\partial w_t}$.*

This deterministic greedy algorithm is usually sub-optimal. For illustration, we present a toy example. We random sample 4 points from the synthetic dataset used by [5] and Section E. The data visualization is shown in Figure 3(a). Given the $w^*$ generated in [5], we perform logistic regression using IMT with GTA for 5 steps. We also perform a brutal-force scan over all possible choices in

each iteration to find the optimal path in the first 5 steps. Train loss versus iterations is plotted in Figure 3(b). All methods are rerun 10 times. We observe that given the optimal path "AAAAA", GTA always chooses "CCCCC".

To tackle the issue, we reformulate Problem 1 to Problem 2 via transformation functions $f, g$. We take linear regression and logistic regression classification as examples and derive corresponding $f, g$ (shown in Appendix B). For other losses, we can derive $f, g$ similarly when they are locally linearized. AIPS introduces randomness to help escape from local optima. As a result, AIPS performs close to the optimal path on average. Here the parameter $c$ is determined by the LSH. In toy example, AIPS chooses among "AAAAA", "AAAAC" and "AACAA", which are near-optimal path. Therefore, the loss of AIPS in each iteration outperforms GTA in Figure 3(b).

**Problem 2** (**Adaptive Inner Product Sampling (AIPS)**). *Given a large set $S$ and a query point $q = g(w^*, w^t)$, and $0 < c < 1$, we aim to sample an example $(x, y) \subset S$ that for any $(x', y') \in S$, $f(x, y)^\top g(w^*, w^t)$ satisfies:*

$$f(x, y)^\top g(w^*, w^t) \geq c \left( f(x', y')^\top g(w^*, w^t) \right). \tag{2}$$

*where $f$ and $g$ are transformation functions with the same dimension.*

### 3.3 Locality Sensitive Sampling as the Key Ingredient

Our algorithm uses LSS as an efficient sampler for Problem 2. Figure 1 shows the complete workflow of the proposed LST. Note that $w_t$ changes in every iteration, and for every iteration, $\mathcal{O}(N)$ computation is required to solve Problem 1. This time complexity is prohibitive in large-scale teaching problems. Therefore, we propose an efficient algorithm to perform IMT by sampling the transformed data vectors that produce large inner products $f(x, y)^\top g(w^*, w^t)$ at the cost of $\mathcal{O}(1)$.

Our algorithm guarantees to find an example for the student model that produces the maximum inner product of $f(x, y)^\top g(w^*, w^t)$ with high probability. The LSS process is designed to sample from a weighted distribution. The probability distribution function in this distribution is a monotonic function of the resulting inner product. We argue that such adaptive sampling should perform much better than random sampling. It is because the probability of selecting the correct sample is $\frac{1}{N}$ for random sampling. On the other hand, for adaptive sampling, this probability is much larger than $\frac{1}{N}$, because by definition, the probability is monotonic to the value of the inner product.

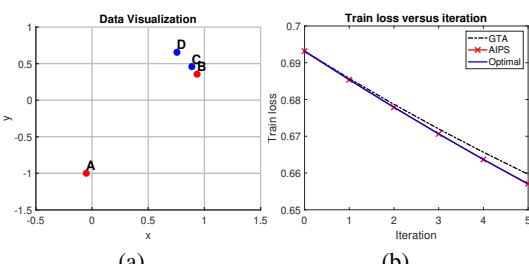

Figure 3: A toy example. (a) visualization of data from 2 classes, red means 0, and blue is 1. (b) Performance after 5 iterations of example path chosen by (1) GTA, (2) AIPS, (3) Optimal.

Moreover, for any monotonic function, the sampling probability distribution changes according to the updates on $w$. Due to monotonicity, the highest probability of choosing the sample which produces the maximum inner product in each iteration is always the highest. The key insight here is that there are two qualities in the inner product, $f(x, y)$ and $g(w^*, w^t)$. With successive iteration, $g(w^*, w^t)$ will change while $f(x, y)$ is always fixed. Therefore, we only need to perform one-time pre-processing to obtain $f(x, y)$ from all $x, y$ pairs in $S$ and put $f(x, y)$ into the LSH tables. Then we can use $g(w^*, w^t)$ as the query for efficient LSS. Then in the following iterations, the hash table data structure remains the same while the query changes to $g(w^*, w^{t+1})$. Therefore, a few hash table lookups are usually sufficient to perform the sampling. The guarantee of the query time is theoretically analyzed in section 4. The detailed sampling process is presented in Algorithm 2. The overall LST algorithm is put in Algorithm 1. In conclusion, after we pay a one-time pre-processing effort of building the hash tables, we need a few or even one lookup in each iteration to obtain a high-quality teaching example, which is far better than a random example and may outperform GTA example.

There are few technical subtleties due to the square of the gradient term. We present it in Equation 1. In Appendix B, we derive the full terms of $f(x, y)$ and $g(w^*, w^t)$ which is similar to a quadratic kernel. Both $f$ and $g$ are the corresponding feature expansion transformation. To increase the

efficiency of LST, we apply asymmetric transforms (details also in Appendix B) to deal with the quadratic terms and demonstrate its validated performance in practice (see experiments in section 6).

| **Algorithm 1:** Locality Sensitive Teaching (LST) | **Algorithm 2:** Locality Sensitive Sampling (LSS) |
|---|---|
| **Result:** Model $w$ 
 **Input:** $D = \{x,\ y\}, w^*, w^t, \eta$; 
 $\pi \leftarrow permute(1, L)$; 
 **for** $i$ *in* $\pi$ **do** 
 $\quad h_1...h_L \leftarrow$ 
 $\qquad H_1(f(x_i, y_i))...H_L(f(x_i, y_i))$; 
 $\quad$ insert $i$ $(id)$ in $L$ hash tables; 
 **end** 
 $l \leftarrow 0$; 
 **while** *not converged* **do** 
 $\quad j \leftarrow$ LSS($f(w^*, w^t)$) (Algorithm 2) ; 
 $\quad w \leftarrow w - \eta(\nabla L(x_j, y_j))$ 
 **end** 
 **return** $w$ | **Result:** Sample $id$ 
 **Input:** Query $q$; 
 $l \leftarrow 0$; 
 $\pi \leftarrow permute(1, L)$; 
 **for** $i$ *in* $\pi$ **do** 
 $\quad$ compute $H(q)$ for hash table $i$; 
 $\quad$ **if** *Bucket B for $H(q)$ is not Empty* **then** 
 $\qquad S \leftarrow$ elements in in $B$; 
 $\qquad id \leftarrow$ random($S$); 
 $\quad$ **else** 
 $\qquad l \leftarrow l + 1$; 
 $\quad$ **end** 
 **end** 
 **return** $id$ |

## 4  Theoretical Insights and Discussions

As shown in section 3.2, we reformulate IMT as an adaptive inner product sampling (AIPS) and provide LST. Here we provide comparison between LST and the current IMT. As shown in Appendix B, GTA used in IMT can be regarded as $\arg\max_i g(w^*, w)^\top f(x_i, y_i)$, which is an exact maximum inner product search (MIPS) problem. [5, 6] only show improvements over SGD, suggesting that GTA is a local optimal solution.

Our AIPS algorithm introduces randomness to GTA. If both $\|f(x_i, y_i)\|$ and $\|g(w^*, w)\|$ are assumed to be the same constant, then from [52], we see GTA and nearest neighbor search (NNS) problem are equivalent, formally,

$$\arg\max_i g(w^*, w)^\top f(x_i, y_i) \approx \arg\min_i \|g(w^*, w) - f(x_i, y_i)\|$$

Instead of doing exact MIPS, AIPS random samples a $c - approximate$ nearest neighbor defined above. The efficiency of AIPS is directly quantified by $\rho = \frac{\log p_1}{\log p_2} < 1$, where $p_1, p_2$ are defined in Section 2.2. The space complexity grows as $O(n^{1+\rho})$, while the query time grows as $O(n^\rho \log n)$, where $n$ is the size of the dataset. Thus, smaller $\rho$ indicates better theoretical performance.

**Theorem 1.** *With a data-structure using functions from $(S_0, cS_0, p_1, p_2)$-sensitive LSH family $\mathcal{H}$, it takes query time $O(n^\rho \log n)$ and space $O(n^{1+\rho})$ to solve AIPS, where $\rho = \frac{\log p_1}{\log p_2}$.*

Theorem 1 shows the time and space complexity for AIPS. In section F, we provide an ablation study of efficiency and accuracy trade-off of our algorithm. Furthermore, asymmetric transformations can be used in accelerating AIPS. A similar trade-off with proofs is shown in [52].

**Theorem 2** (**Exponential Teachability of LST**). *Following the theoretical setting in the synthesis-based teaching [5], the teacher is assumed to be able to provide any examples from*

$$\mathcal{X} = \{x \in \mathbb{R}^d, \|x\| \le R\},$$
$$\mathcal{Y} = \{-1, 1\} \text{ (Classification) or } \mathbb{R} \text{ (Regression)}$$

*Let $\eta \ne 0$ denote the fixed learning rate of the student. Let $\ell(\cdot, \cdot)$ denote the loss function. Moreover, $\ell(\cdot, \cdot)$ has the following propriety: for any $w \in \mathbb{R}^d$, there exists a $\gamma$ with $\gamma \ne 0$ and $\|\gamma\| \le \frac{R}{\|w - w^*\|}$ that, if $\hat{x} = \gamma(w - w^*)$ and $\hat{y} \in \mathcal{Y}$, we have*

$$0 < \gamma \nabla_{\langle w, \hat{x} \rangle} \ell(\langle w, \hat{x} \rangle, \hat{y}) \le \frac{1}{\eta}. \tag{3}$$

*Suppose for a $(S_0, cS_0, p_1, p_2)$-sensitive hashing function family $\mathcal{H}$ where $S_0 = \max_i f(\hat{x}, \hat{y})^\top g(w^*, w^i)$ and*

$$\hat{x} = \gamma\left(w^t - w^*\right) \text{ and } \hat{y} \in \mathcal{Y}, \tag{4}$$

*then LST can achieve exponential teachability with probability at least $p_1$ in a single iteration.*

Theorem 2 shows the condition and probability of LST to achieve exponential teachability in each iteration. There is a probability $p_1$ to perfectly imitate the omniscient teacher. More importantly, it introduces some randomness into the teaching algorithm and therefore makes the teaching examples more diverse and plausible for humans. Note that, all Lipschitz smooth and strongly convex loss functions can satisfy Equation 3. Using Theorem 2, we can easily extend LST's exponential teachability to the other teaching scenarios such as combination-based teaching [5], (rescaled) pool-based teaching [5] and black-box teaching [6]. Detailed proofs are shown in Appendix C.

**Time Complexity Analysis**: Theorem 1 and 2 provide the analysis for the LSS algorithm to achieve the exponential teachability with provably sub-linear time complexity. In practice, we need to pay one-time pre-processing time $\mathcal{O}(NKL)$ to build the hash tables, where $N$ is the number of samples, $K$ is the number of hash functions in each hash table, and $L$ is the number of hash tables. During inference, we usually only use a few lookups (typically just one), *i.e.*, $\mathcal{O}(1)$ complexity, in each iteration to get a high-quality teaching example. We also conduct an empirical experiment to show the dedicated trade-off between the accuracy and efficiency using different $K$ and $L$. Therefore, LST reduces the time complexity for each teaching iteration from $\mathcal{O}(N)$ to $\mathcal{O}(1)$ by hash table lookups.

## 5   Locality Sensitive Teaching on Integrated CPU-GPU SoC

In this section, we present the LST system implemented on an integrated CPU-GPU SoC platform. In recent machine learning based IoT applications such as autonomous driving, smart home, and intelligent robots, devices with integrated CPU-GPU architecture (e.g., NVIDIA's Jetson [54] series) is gaining increasing preference. The smart co-optimization[55, 56] of machine learning algorithms on CPU and GPU resources determines their overall performance on IoT deployment. In this work, our system design is based on two observations: (1) The CPU resources could be more utilized on the integrated CPU-GPU SoC. For instance, a dual-core NVIDIA Denver 2 64-bit CPU and a quad-core ARM Cortex-A57 are equipped on NVIDIA TX2 [57] SoC. Vision-based machine learning applications such as object detection [55] do not regard on-chip CPU as a computation resource. In our problem, if we can migrate some of the computation to those powerful CPUs, the overall efficiency can be improved. (2) Hashing is cheap on CPU. Most GPUs suffer memory constraints. Although GPU-based LSH hash tables are 1.5x faster over CPU [58] in nearest neighbor search, it stores the hash table entirely on GPU memory. As a result, the availability of GPU for other machine learning operations is limited. To tackle the issue, hybrid CPU-GPU LSH systems [58, 59] become powerful in both academics and industry.

In our LST systems on integrated CPU-GPU SoC, we implement LSH by separating the random projection and hash table lookups into GPU and CPU. We first generate hash codes of data vectors by GPU-based random matrix multiplication via CuPy [60] and compiled CUDA kernels. Then, we insert the hash codes to hash tables built on the CPU. The query vector is transformed into hash codes in the query phase using the same random matrix in GPU. Then, the query hash codes are sent to the CPU and perform hash table lookups. We provide Cython wrapping for the implementation to make it PyTorch friendly. We believe this implementation would be more beneficial for the community as it can be easily plugged into any deep learning model.

## 6   Experiments

In this section, we experiment the effectiveness and efficiency of LST in various scenarios. We first perform the algorithmic level evaluation of LST on different datasets. There are three questions that we would like to answer: (1) does the LST performs better or the same compared to the current IMT in teaching the learner model towards faster convergence in iteration-wise? (2) compared to the IMT, does the LST algorithm reduce the computation time when achieving the same performance? (3) is the proposed LST algorithm scalable in the dataset that the IMT is almost infeasible in practice? Next, we conduct experiments to demonstrate the scalability of LST on IoT devices. There are two questions that we would like to answer: (1) does the LST's teachability robust to IoT devices? (2) compared to IMT, is the LST energy efficient on IoT devices? Note that we present the results of regression in this section. For classification results, please refer to Appendix E.

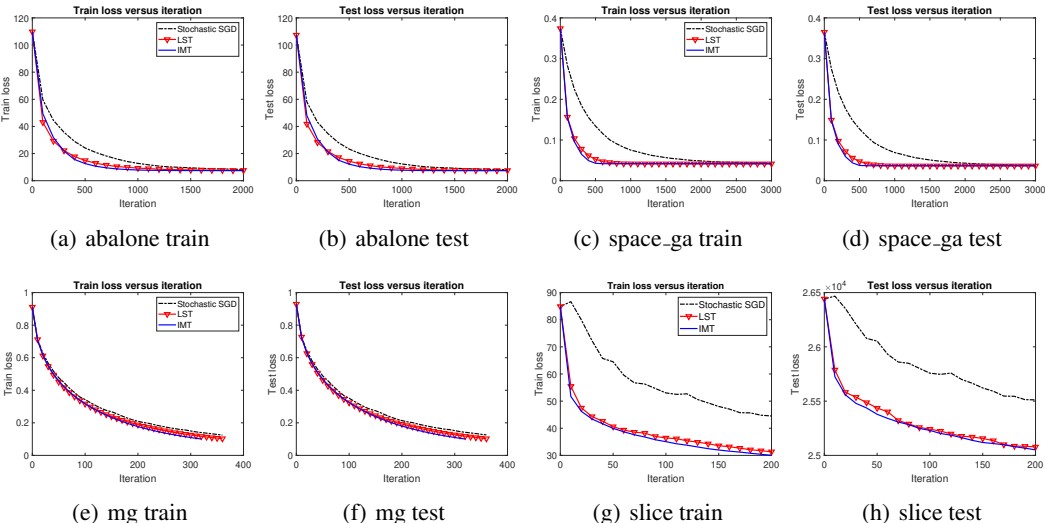

| (a) abalone train | (b) abalone test | (c) space_ga train | (d) space_ga test |

| (e) mg train | (f) mg test | (g) slice train | (h) slice test |

Figure 4: Average train and test loss versus iteration of: abalone in (a) and (b), space_ga in (c) and (d), mg in (e) and (f), slice in (g) and (h). Results have demonstrated that both LST and the current IMT are superior to stochastic SGD in guiding the learner model towards faster convergence. Meanwhile, LST has a similar performance with IMT as their train and test losses descend almost the same way.

**Datasets.** In the experiments, we use four regression datasets to demonstrate the performance of our LST. First, we use abalone, space_ga [61] and mg dataset from LIBSVM [61] and UCI dataset [62]. The abalone dataset contains 4177 8-dimensional feature vectors. The space_ga and mg datasets contain 3107 and 1385 7-dimensional vectors, respectively. Then we test LST's performance on large scale regression datasets to validate its scalability. We use slice dataset from UCI dataset [62]. slice dataset contains 53500 training samples and 42800 testing samples. Each sample is a 74-dimensional vector. We use slice only for algorithm level evaluation as it causes memory exhaustion on IoT devices. We randomly split 30% of samples in each dataset as a test set while others are training set. All datasets are under MIT license.

### 6.1 Evaluation on Locality Sensitive Teaching

In this section, we present the performance of our proposed LST algorithm in teaching regression models towards fast convergence. The evaluation is on a server with 1 Nvidia Tesla V100 GPU and two 20-core/40-thread processors (Intel Xeon(R) E5-2698 v4 2.20GHz). Given the four regression datasets, we set the learner model as a linear regression model with Mean Square Error (MSE) loss to compare the performance of LST, IMT, and stochastic SGD. Here, we consider the stochastic SGD as a random teaching approach. In each step, the three algorithms above choose an example from the dataset and feed it into the learner. The learner model then processes the example and performs AdaGrad[63] to update the weights accordingly. We use early stopping on the learner model to prevent overfitting. Details of LST for classification are in Appendix E.

**Convergence.** In Figure 4, we plot the train/test loss versus iteration for three teaching approaches mentioned above on the 4 regression dataset. From the (a), (b), (c), and (d), we observe that the LST and IMT [5] algorithm teaches the learner models that converge to optimum with much less iterations than stochastic SGD. This performance shows that both LST and IMT leads to faster iteration-wise convergence of learner models in train and test set. Moreover, in each step, the learners' train and test losses taught by LST are close or even better than IMT. In sub-figure (c) and (d), we show that LST also approximates IMT in teaching towards faster convergence even when the gap of train losses and test losses between IMT and stochastic SGD are small. This result demonstrates that LST's approximation error to the optimal path is affordable even in some settings that IMT only has few advantages towards stochastic SGD. In sub-figure (e) and (f), we demonstrate that LST and IMT share similar performance in large scale regression sets (*e.g.* slice) on average. Moreover, we observe that about 30% LST instances outperform IMT. The effectiveness experiment answers the first question. LST can achieve the same fast convergence effect in iteration-wise when compared

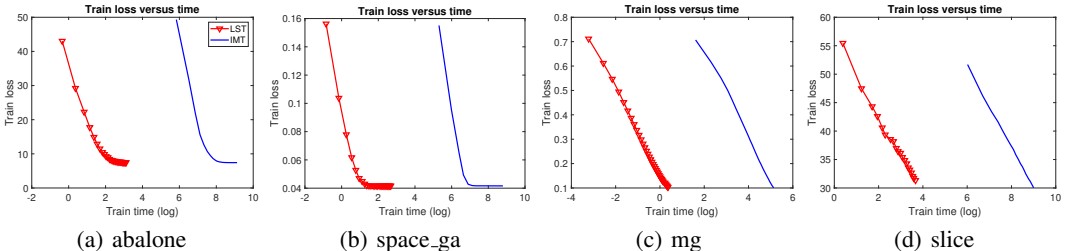

| (a) abalone | (b) space_ga | (c) mg | (d) slice |

Figure 5: Average train loss versus the train time (seconds in log scale) of abalone in (a), space_ga in (b), mg in (c), slice in (d). The time unit is second. Results have shown that LST is 2000x times faster than the current IMT to achieve the same loss at each step.

to IMT. With non-negligible probability, LST's teachability outperforms IMT in teaching regression towards convergence with fewer iterations.

**Speed.** In Figure 5, we plot the train loss versus train time (seconds in log scale) for LST and IMT in four regression sets. In each iteration, we record the time for teaching algorithms to select an example and the weight update time for the learners given this example. In the four sub-figures, we observe a consistent gap between two approaches in time. At the same loss level during training, our LST teaches 2000× faster than IMT on average. For the large scale slice dataset, it takes 404.95 seconds on average for IMT to finish one iteration. This long-time prevents IMT from online deployment at scale. Our LST tackles the issue by reducing the time to 0.196 seconds while preserving the same performance in iteration-wise. This significant acceleration makes LST scalable in large scale teaching scenarios. This experiment answers the second and third questions. LST is 2000× faster than IMT while achieving the same iteration-wise fast convergence. Moreover, LST is efficient in teaching when performed on large scale datasets. We also perform a parameter study in Appendix F.

## 6.2 Scalability on IoT Devices

In this section, we compare LST and IMT on Nvidia TX2 devices. For detailed settings, we refer to Appendix G. We first observe that the iteration-wise performance remains identical to the evaluation on the server. This phenomenon answers the first question: LST's teachability is robust in IoT devices by achieving exactly the same iteration-wise conver-

| Dataset | Energy Savings | Speedups |
|---------|----------------|----------|
| abalone | 99.76% | 425.12× |
| space_ga | 99.34% | 149.07× |
| mg | 98.73% | 117.20× |

Table 1: Efficiency of LST over IMT on TX2

gence as server-based experiments. Then, in Table 1, we present the energy savings and speedups of LST over IMT on four datasets. According to the results, we observe that LST achieves at least 98.73% energy savings and 117.20× speedups over IMT. In the synthetic classification dataset, LST is 306.12× faster than IMT with 99.67% energy savings. This performance gain is fewer than server-based experiments due to the less available resources on TX2 SoCs (e.g., 105W power budget CPU in sever setting vs. 7.5W power budget in IoT setting). However, this improvement is still promising and answer the second question: LST can gain at most 99.76% energy savings with over 100× speedups compared with IMT on IoT devices.

## 7 Concluding Remarks

The inefficiency of iterative machine teaching (IMT) prevents it from being used in IoT-based teaching scenarios at home. There is no sub-linear time iterative teaching algorithm that enables teaching in large-scale real-world settings before our work. The paper proposes a practical and fast teaching approach. By reformulating IMT's optimization problem to an adaptive sampling problem, we propose a Locality Sensitive Teaching (LST) algorithm. LST achieves exponential teachability and reduces the time complexity of each iteration from $\mathcal{O}(N)$, where $N$ is the number of teaching examples, to few $\mathcal{O}(1)$ lookups in hash tables. Moreover, we design an LST system that fully exploits the resources of integrated CPU-GPU SoCs on IoT devices. We demonstrate, both theoretically and empirically, LST's teachability improvements, speed acceleration, and tremendous energy reductions

over IMT during experiments on IoT devices. The development of LST makes teaching algorithms feasible on more on-device personalized education applications.

## Acknowledgements

Zhaozhuo Xu and Anshumali Shrivastava are supported by the National Science Foundation IIS-1652131, BIGDATA-1838177, AFOSR-YIP FA9550-18-1-0152, the ONR DURIP Grant, and the ONR BRC grant on Randomized Numerical Linear Algebra. Chaojian Li and Yingyan Lin are supported by the National Science Foundation EPCN under No. 1934767. Weiyang Liu is supported by a Cambridge-Tübingen Fellowship, an NVIDIA GPU grant, DeepMind and the Leverhulme Trust via CFI. We would like to thank Tracy Volz and Elizabeth Festa for great discussion on scientific writing.

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
