# Appendix

## A  Locality Sensitive Hashing

In this paper, we define $\mathcal{H}$ as a LSH function family. For every $h \in \mathcal{H}$ and $h : \mathbb{R}^D \to \{0, 1\}$, we show that:

**Definition 1** (**LSH Family**). *We say a family $\mathcal{H}$ is $(S, cS, P_1, P_2)$-sensitive if it satisfies: given any $x, y \in \mathbb{R}^D$, for any $h$ that uniformly sample from $\mathcal{H}$, we have:*

- *if $Sim(x, y) \geq S$ then $Pr(h(x) = h(y)) \geq P_1$*
- *if $Sim(x, y) \leq cS$ then $Pr(h(x) = h(y)) \leq P_2$*

Here, we denote $Sim : \mathbb{R}^D \times \mathbb{R}^D \to \mathbb{R}$ as a similarity function. $P_1, P_2$ are two probabilities with value between 0 and 1 and $P_1 > P_2$. In practice, we would use $K \times L$ LSH functions to build $L$ hash tables. Each table has $K$ independent LSH functions. In the query phase, given a query vector, we first generate $L$ $K$-bit hash values using the LSH functions. Then, we lookup the items that has the same hash values with the query in at least one table. For details about LSH, we refer the readers to [12, 51, 48].

In practice, a function $h$ in the LSH family $\mathcal{H}$ for similarity $Sim$ must satisfies the following condition:

$$Pr(h(x) = h(y)) = f(Sim(x, y)), \tag{5}$$

where function f is monotonically increasing.

**Algorithm and Implementation Details**  Given a right LSH function $h$, its collision probability is $cp(x, y) = Pr(h(x) = h(y))$. For both regression and classification, we use SimHash [64]. We first normalize the data. Then collision probability of SimHash is $cp(x, y) = 1 - \frac{cos^{-1}(\frac{x \cdot y}{||x||_2 ||y||_2})}{\pi}$. This collision probability is monotonic in the inner product. Details of Simhash are shown in [12, 51, 48]. For LSH based AIPS, once we set the parameters $(K, L)$, the constant $c$ in Problem 2 is determined. Furthermore, we centered the data stored in the LSH to make the query more efficient.

## B  Detailed Derivation

In this section, we present the derivation from GTA objective to an adaptive inner product sampling problem as discussed in 3.2. We introduce formula for both the regression and classification.

### B.1  Regression

#### B.1.1  Loss and Gradient

The GTA objective is:

$$x, y = \arg \min_{x,y} \tag{6}$$

$$\eta^2 \left\| \frac{\partial \ell(\langle w^t, x \rangle, y)}{\partial w_t} \right\|_2^2 - 2\eta \left\langle w^t - w^*, \frac{\partial \ell(\langle w^t, x \rangle, y)}{\partial w^t} \right\rangle$$

For the linear regression model with MSE loss, we have

$$\frac{\partial \ell(\langle w^t, x \rangle, y)}{\partial w^t} = 2(w^T x - y)x = 2(-y + \sum_1^n w_i x_i)x \tag{7}$$

Thus, the right term can be decomposed as:

$$
x, y = \arg\min_{x,y} \begin{pmatrix} w_1^t w_1^t \\ w_1^t w_2^t \\ \vdots \\ w_d^t w_d^t \\ -2w_1^t \\ \vdots \\ -2w_d^t \\ 1 \\ -w_1^t(w_1^t - w_1^*) \\ -w_1^t(w_2^t - w_2^*) \\ \vdots \\ -w_d^t(w_d^t - w_d^*) \\ w_1^t - w_1^* \\ \vdots \\ w_d^t - w_d^* \end{pmatrix}^T \begin{pmatrix} \eta\|x\|^2 x_1 x_1 \\ \eta\|x\|^2 x_1 x_2 \\ \vdots \\ \eta\|x\|^2 x_d x_d \\ \eta\|x\|^2 x_1 y \\ \vdots \\ \eta\|x\|^2 x_d y \\ \eta\|x\|^2 y^2 \\ x_1 x_1 \\ x_1 x_2 \\ \vdots \\ x_d x_d \\ x_1 y \\ \vdots \\ x_n y \end{pmatrix} \tag{8}
$$

After normalizing $\|x\|_2 = 1$, we transform the GTA optimization problem of IMT as an adaptive inner product sampling problem with $k = 1$, which can also be interpreted as a maximum inner product problem $x, y = \arg\max f(x,y)^\top g(w^*, w)$, where $f$ and $g$ are:

$$
f(x,y) = \begin{pmatrix} x_1 x_1 \\ x_1 x_2 \\ \vdots \\ x_d x_d \\ x_1 y \\ \vdots \\ x_d y \\ y^2 \end{pmatrix} \qquad g(w^*, w) = - \begin{pmatrix} (\eta - 1)w_1^t w_1^t + w_1^t w_1^* \\ (\eta - 1)w_1^t w_2^t + w_1^t w_2^* \\ \vdots \\ (\eta - 1)w_d^t w_d^t + w_d^t w_d^* \\ (1 - 2\eta)w_1^t - w_1^* \\ \vdots \\ (1 - 2\eta)w_d^t - w_d^* \\ \eta \end{pmatrix} \tag{9}
$$

In practice, building $f$ and $g$ requires the computation for $w_i^t w_j^t$ for all $i, j$. To accelerate the computation, we can only compute $w_i^t w_i^t$ and multiply it by $d$ times, where $d$ is the dimension of $x$ and $w^t$. In this way, we still get good empirical results as shown in section 6.

## B.2 Classification

For the classification task with the logistic regression model, we modify the formula of logistic regression in teaching objectives to make it convenient for derivation. Then, we reformulate the optimization problem in IMT by an adaptive inner product sampling problem.

### B.2.1 Loss and Gradient

The sigmoid function:

$$
\sigma(s) = \frac{e^s}{1 + e^s} = \frac{1}{1 + e^{-s}} \tag{10}
$$

Thus,

$$
P(y|x) = \sigma(y\theta^T x) \tag{11}
$$

Recall that the probability of getting the $y_1, ..., y_m$ from the corresponding $x_1, ..., x_m$:

$$P(y_1, ...y_m|x_1...x_m) = \prod_{i=1}^{m} P(y_i|x_i) \qquad (12)$$

We modify the loss as a maximum likelihood loss, specifically, we want to maximize the likelihood,

$$\max \prod_{i=1}^{m} P(y_i|x_i) \leftrightarrow \max ln(\prod_{i=1}^{m} P(y_i|x_i))$$

$$= \max \sum_{i=1}^{m} lnP(y_i|x_i)$$

$$\leftrightarrow \min(-\frac{1}{m} \sum_{i=1}^{m} lnP(y_i|x_i)) \qquad (13)$$

$$= \frac{1}{m} \sum_{i=1}^{m} ln\frac{1}{P(y_i|x_i)}$$

$$= \frac{1}{m} \sum_{i=1}^{m} ln\frac{1}{\sigma(y_i w^T x_i)}$$

$$= \min \frac{1}{m} \sum_{i=1}^{m} ln(1 + e^{-y_i w^T x_i})$$

Thus

$$\frac{\partial \ell(\langle w^t, x \rangle, y)}{\partial w_t} = -\frac{xy}{1 + exp(-w_t^T xy)} \qquad (14)$$

$$\left\| \frac{\partial \ell(\langle w^t, x \rangle, y)}{\partial w_t} \right\|_2 = \frac{1}{1 + exp(-w_t^T xy)} \qquad (15)$$

### B.2.2 Formulation

The GTA objective is:

$$x, y = \arg\min_{x,y} \quad \eta^2 \left\| \frac{\partial \ell(\langle w^t, x \rangle, y)}{\partial w_t} \right\|_2^2 \qquad (16)$$

$$- 2\eta \left\langle w^t - w^*, \frac{\partial \ell(\langle w^t, x \rangle, y)}{\partial w^t} \right\rangle \qquad (17)$$

we have

$$\frac{\partial \ell(\langle w^t, x \rangle, y)}{\partial w^t} = -\frac{xy}{1 + exp(-w^{t\,T} xy)}$$

$$\left\| \frac{\partial \ell(\langle w^t, x \rangle, y)}{\partial w^t} \right\|_2 = \frac{1}{1 + exp(-w^{t\,T} xy)}$$

Therefore,

$$x, y = \arg\min_{x,y} \quad \eta^2 \left\| \frac{\partial \ell(\langle w^t, x \rangle, y)}{\partial w^t} \right\|_2^2 - 2\eta \left\langle w^t - w^*, \frac{\partial \ell(\langle w^t, x \rangle, y)}{\partial w^t} \right\rangle$$

$$= \arg\min_{x,y} \quad \frac{\eta}{(1 + exp(-w^{t\,T} xy))^2} + \frac{2(w^t - w^*)^T xy}{1 + exp(-w^{t\,T} xy)} \qquad (18)$$

let $z = xy$, we can approximate the solution of the problem above by finding

$$x, y = \arg \max_{z,\ (x,y) \in z} (w^* - w^t)^T z \tag{19}$$

Here, we reformulate the formula by finding the maximum inner product between $g(w^*, w) = w^* - w^t$ and $f(x, y) = z = xy$, which is an adaptive inner product sampling scheme. Empirically, it achieves promising results as shown in Figure 6 and 7 in section 6.

## C  Proofs

### C.1  Theorem 2

In iteration $t$, the omniscient teacher chose one teaching example by solving the following optimization problem

$$\min_{x \in \mathcal{X}, y \in \mathcal{Y}} \eta^2 \| \nabla_{w^t} \ell \left( \langle w^t, x \rangle, y \right) \|^2 - 2\eta \left\langle w^t - w^*, \nabla_{w^t} \ell \left( \langle w^t, x \rangle, y \right) \right\rangle.$$

which can be reduces to the following optimization problem:

$$x, y = \arg \max_{(x,y) \in S} f(x,y)^\top g(w^*, w^t).$$

Suppose for a $(S_0, cS_0, p_1, p_2)$-sensitive hashing function family $\mathcal{H}$ where we have

$$S_0 = \max_i f(\hat{x}, \hat{y})^\top g(w^*, w^i),$$

and

$$\hat{x} = \gamma \left( w^t - w^* \right) \text{ and } \hat{y} \in \mathcal{Y},$$

then we have that for any $(x', y') \in S$, if $f(x', y')^\top g(w^*, w^t) \geq S_0$

$$Pr \left( h(f(x', y')) = h(g(w^*, w^t)) \right) \geq p_1, \tag{20}$$

where $h$ is some hash function. This means that with probability at least $p_1$, a example $(x_s, y_s)$ sampled by this hash function satisfies the following (we assume there must exist some $(x, y)$ such that $f(x, y)^\top g(w^*, w^t) \geq S_0$):

$$f(x_s, y_s)^\top g(w^*, w^t) \geq \max_i f(\hat{x}, \hat{y})^\top g(w^*, w^i) \geq f(\hat{x}, \hat{y})^\top g(w^*, w^t). \tag{21}$$

By transforming Eq. 21 into the GTA, we have that

$$\begin{aligned}
\min_{x \in \mathcal{X}, y \in \mathcal{Y}} & \eta^2 \| \nabla_{w^t} \ell \left( \langle w^t, x \rangle, y \right) \|^2 - 2\eta \left\langle w^t - w^*, \nabla_{w^t} \ell \left( \langle w^t, x \rangle, y \right) \right\rangle \\
\leq & \eta^2 \| \nabla_{w^t} \ell \left( \langle w^t, x_s \rangle, y_s \right) \|^2 - 2\eta \left\langle w^t - w^*, \nabla_{w^t} \ell \left( \langle w^t, x_s \rangle, y_s \right) \right\rangle \\
\leq & \left( \eta^2 \beta^2_{(\langle w^t, \hat{x} \rangle, \hat{y})} \gamma^2 - 2\eta \beta_{(\langle w^t, \hat{x} \rangle, \hat{y})} \gamma \right) \| w^t - w^* \|_2^2,
\end{aligned} \tag{22}$$

where we denote $\beta_{(\langle w, x \rangle, y)}$ as the gradient $\nabla_{\langle w, x \rangle} \ell \left( \langle w, x \rangle, y \right)$ with respect to $\ell(\cdot, \cdot)$. Plug Eq. (22) into the following recursion:

$$\begin{aligned}
\left\| w^{t+1} - w^* \right\|_2^2 &= \left\| w^t - \eta \frac{\partial \ell(\langle w, x \rangle, y)}{\partial w} - w^* \right\|_2^2 \\
&= \left\| w^t - w^* \right\|_2^2 + \eta^2 \left\| \frac{\partial \ell(\langle w^t, x \rangle, y)}{\partial w^t} \right\|_2^2 \\
&\quad - 2\eta \left\langle w^t - w^*, \frac{\partial \ell(\langle w^t, x \rangle, y)}{\partial w^t} \right\rangle.
\end{aligned} \tag{23}$$

Then we have that

$$\begin{aligned}
\left\|w^{t+1}-w^*\right\|_2^2 &= \min_{x\in\mathcal{X},y\in\mathcal{Y}}\left\|w^t-\eta\frac{\partial\ell(\langle w,x\rangle,y)}{\partial w}-w^*\right\|_2^2 \\
&= \left\|w^t-w^*\right\|_2^2 + \min_{x\in\mathcal{X},y\in\mathcal{Y}}\eta^2\left\|\frac{\partial\ell(\langle w^t,x\rangle,y)}{\partial w^t}\right\|_2^2 \\
&\quad - 2\eta\left\langle w^t-w^*,\frac{\partial\ell(\langle w^t,x\rangle,y)}{\partial w^t}\right\rangle \\
&\leq \left(1+\eta^2\beta_{(\langle w^t,\hat{x}\rangle,\hat{y})}^2\gamma^2-2\eta\beta_{(\langle w^t,\hat{x}\rangle,\hat{y})}\gamma\right)\left\|w^t-w^*\right\|_2^2 \\
&= \left(1-\eta\beta_{(\langle w^t,\gamma(w^t-w^*)\rangle,\hat{y})}\gamma\right)^2\left\|w^t-w^*\right\|_2^2.
\end{aligned} \tag{24}$$

First we let $\nu(\gamma)=\min_{w,y}\gamma\nabla_{\langle w,\gamma(w-w^*)\rangle}\ell\left(\langle w,\gamma(w-w^*)\rangle,y\right)$.

Then we know that $0<\nu(\gamma)\leq\gamma\beta_{(\langle w,\gamma(w-w^*)\rangle,\hat{y})}\leq\frac{1}{\eta}<\infty$ for any $w,y$,

Following the previous steps, we now have

$$0\leq 1-\gamma\eta\beta_{(\langle w,\gamma(w-w^*)\rangle,\hat{y})}\leq 1-\eta\nu(\gamma),$$

Next, we perform a simplification of $\nu(\gamma)$ to $\nu$.

Next, using Eq. (24), we have:

$$\left\|w^{t+1}-w^*\right\|_2^2\leq(1-\eta\nu)^2\left\|w^t-w^*\right\|_2^2,$$

Finally, we show that we could obtain the exponential convergence:

$$\left\|w^t-w^*\right\|_2\leq(1-\eta\nu)^t\left\|w^0-w^*\right\|_2,$$

The student needs $\left(\log\frac{1}{1-\eta\nu}\right)^{-1}\log\frac{\|w^0-w^*\|}{\epsilon}$ samples to approximate $w^*$ with at most $\epsilon$ in Euclidean distance. It also indicates that with probability at least $p_1$, the LST teacher can achieve exponential teachability in the iteration $t$. In order to achieve exponential teachiability in $T$ iterations, the sufficient condition in Eq. (22) must be satisfied in all $T$ iterations. The probability is at least $p_1^T$.

## D    Motivating Example Settings

In the toy example shown in Section 1, we introduce a teaching task that helps students identifying COVID-19 CT images. We use the COVID-CT [53] dataset containing COVID-19 or regular CTs that confirmed by the clinic. We random sample 50 positive examples and 50 negative samples. Then, we use a pre-trained DenseNet [65] shown in [53] to generate 1024 dim features and the confidence score for each image. After then, we regard the weights in the last layer of DenseNet as $w^*$ shown in Section 3.2. Finally, we initialize a linear model with random weights and perform LST and IMT by feeding a feature-score pair from a dataset image. We can observe LST's superiority over IMT in the time wise convergence.

## E    Locality Sensitive Teaching for Classification

We also present experiments of our LST on two classification datasets. The first classification dataset is a synthetic dataset used in [5]. Specifically, we combine two Gaussian distribution centered in (0.6,0.6) (label 1) and (-0.6,-0.6) (label -1) to form a dataset. This dataset is used for observing the training behavior, and therefore, no test set is prepared. The second classification dataset is the ALOI [66] dataset. We use its LIBSVM [61] version, which contains 108,000 samples labeled in 1000 classes. Each sample has 128 attributes. Following [67], we trim the ALOI dataset as a binary classification set by grouping classes 1,2 and 3,4,...1000. We use ALOI only for algorithm level evaluation as it causes memory exhaustion on IoT devices. We randomly split 30% of samples

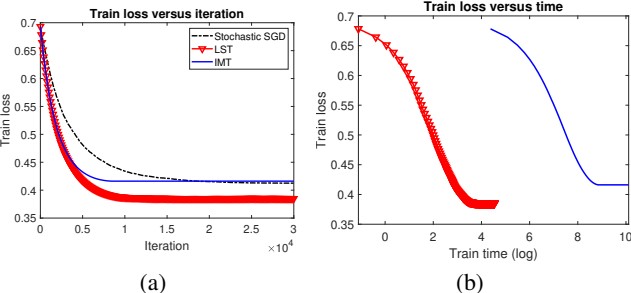

(a)        (b)

Figure 6: (a) Train loss versus iteration in synthetic dataset. (b) Train loss versus time in synthetic dataset. LST outperforms IMT and Stocastic SGD in leading towards fast and better convergence.

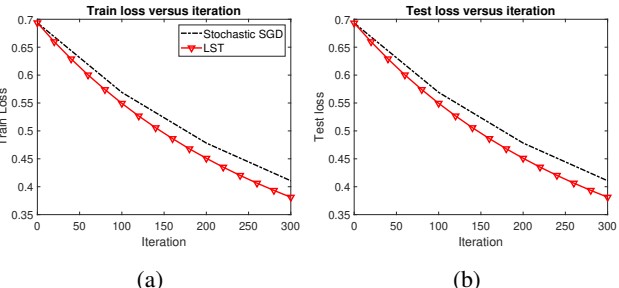

(a)        (b)

Figure 7: Train loss (a) or test loss (b) versus iteration in ALOI dataset. IMT is infeasible in this dataset for the unaffordable time spent in selecting teaching examples. But our LST performs teaching via adaptive inner product sampling.

in ALOI as test set while others form the train set. We first present the performance of the LST algorithm in teaching classification models towards fast convergence. We perform logistic regression with cross-entropy loss to compare the performance of LST, IMT, and stochastic SGD. In each step, the three algorithms above feed a sample into the learner model. The learner model then performs gradient descent to update the weights.

**Synthetic dataset.** In Figure 6, we plot the curves of train loss versus iteration and train loss versus time of the learner models taught by LST, the IMT, and stochastic SGD. From (a), we observe that compared to stochastic SGD both LST and the current IMT algorithm teach the learner model to converge faster in iteration. The learner model taught by the current IMT and stochastic SGD converges to the same loss level while LST's learner convergences to a lower loss than them. From (b), we observe that LST teaches the learner model $2000\times$ faster than the IMT to achieve the same loss during training. Also, similar to (a), LST's learner model converges to a lower loss, indicating better optimization performance.

These results provide a case to answer the first and second questions. 1. LST is more effective on average for it avoids the local minimum and achieves a lower loss level in both train and test set. 2. Compared with IMT, LST can achieve teaching effect with $2000\times$ acceleration in teaching speed during each iteration.

**ALOI dataset.** In the ALOI dataset, the IMT is infeasible due to the memory exhaustion in Nvidia V100 GPU. Given the conditions that V100 has 32G memory, it is shown that the IMT algorithm may be infeasible when training on this real classification dataset via a single GPU. However, our LST still works in a fast convergence task. In Figure 7, we plot: (a) the train loss versus iteration, (b) the test loss versus iteration, of the learner (logistic regression model) taught by three teaching approaches. From (a) and (b), we observe that LST's learner achieves lower train or test loss than stochastic SGD in each iteration, indicating its teachability. This classification experiment answers the third question partially. When the IMT is unscalable and infeasible in some real-world datasets due to the large time and space complexity. LST still accomplishes the iterative teaching procedure and outperforms stochastic SGD with lower train and test loss in each iteration.

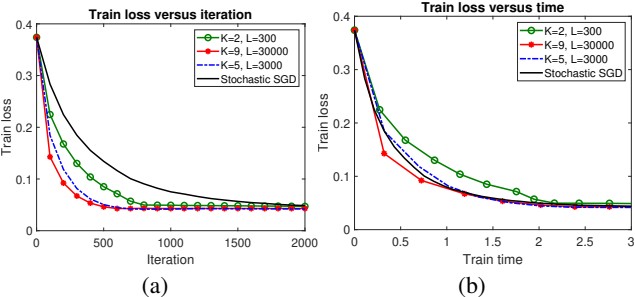

Figure 8: (a) Train loss versus iteration, (b) train loss versus time (in seconds), of LST in space_ga dataset with different $K$ and $L$.

# F  Parameter Study

In this section, we perform a study on two parameters of our LST. The number of random projections $K$ and the number of tables $L$. In Figure 8, we plot the training loss versus iteration and time in space_ga dataset. From (a), we observe that $K = 2$ and $L = 300$ is enough for LST to outperform stochastic SGD in iteration-wise convergence. Also, as we increase the $K$ and $L$, the performance of LST becomes better. This effect is reasonable as more random projections may introduce a lower approximation error. From (b), we observe that, with the appropriate $K$ and $L$, LST converges with less time than stochastic SGD. Meanwhile, lower $K$ or $L$ may lead to less time-wise efficiency than stochastic SGD due to the performance drop in iteration-wise convergence.

The parameter study provides an answer to the two questions. First, appropriate $K$ and $L$ outperforms stochastic SGD in iteration-wise and time-wise convergence. Second, different $K$ and $L$ changes the performance of LST, but the optimal parameter remains stable and robust over different teaching datasets.

# G  IoT Settings

For evaluation, NVIDIA TX2 [57], a device with a SoC consisting of a 256-core Pascal GPU and 6-cores CPU targeting IoT applications [68, 69] is used as the platform. Following the hardware configuration in [69], we pre-set NVIDIA TX2 in the *max-N* mode to make full use of the computing resource. Specifically, the measured energy and latency come from the output of sysfs [70] of the embedded INA3221 [71] power rails monitor in NVIDIA TX2.