# OpenReview forum: "Locality Sensitive Teaching"
_NeurIPS.cc/2021/Conference — NeurIPS 2021 Poster_

### Official Review · Reviewer_sssx · 2021-07-14

**Rating:** 6
**Confidence:** 3

**Summary:**

This paper considers an iterative machine learning task of how an omniscient teacher choosing learning data optimally such that the student can learn fast from the data via gradient descent. It proposes an adaptive inner product sampling (AIPS) algorithm that uses locality-sensitive hashing-based sampling to make the search much more efficient than greedy approaches. The authors show the exponential teachability of their approach and provide empirical studies.

**Limitations And Societal Impact:**

The authors addressed them.

**Main Review:**

The main novelty of the work is to transform the iterative machine learning problem to an adaptive inner product sampling problem with transform functions f and g. Combined with hashing structure, the proposed algorithm only need to traverse the dataset in pre-processing step to calculate a feature representation $f(x_i,y_i)$ for each data point $(x_i,y_i)$, which enables the exponential teachability.

Major question/concern:
1.  When you say "reformulate problem 1 to problem 2" at Line 158, does that mean there always exists transformation functions f and g such that problem 2 is equivalent to problem 1? Is there an explicit way to construct f and g for general loss function l apart from MSE and logistic regression?
2.  Presentation of the paper is incomplete, please define notations before using them even they are conventional, especially for key results. E.g., what does the notation permute(1, L) and function H(\cdot) mean in Algorithm 2? What does $(S_0,cS_0,p_1,p_2)$-sensitive hashing function family mean in Theorem 2.
3.  The test error for slice test in the plot (h) of figure 4 is way higher than the training error, why?

Minor comments:
1.  It's not a good idea to mix related work and preliminaries, as preliminaries should give clear definitions and results as a basement for the work. Once you mix them it's hard for readers to find such information and the way you present them is not clear and rigorous.
2.  You should link the reference by using cite, and ref the equations using eqref, e.g. in line 205.
3.  The order of figures 1,2,3 is inappropriate, as you mention figure 2,3 before figure 1.
4. Expressions like "Following the theoretical setting in the synthesis-based teaching [5]" should not show up in a Theorem. You can either explicitly state the setting outside the theorem and ref it in theorem, or explicitly state the setting inside the theorem. Either way, settings and assumptions should be explicitly stated in a rigorous way even it's brought somewhere else.

**Time Spent Reviewing:**

4

---

> ### Author Response · Authors · 2021-08-10
> **Thank you for the comprehensive review, please see the following clarifications.**
>
> Thanks for your suggestions to help us improve the paper!
>
> **Q1: Does that mean there always exists transformation functions f and g such that problem 2 is equivalent to problem 1? Is there an explicit way to construct f and g for general loss function l apart from MSE and logistic regression?**
>
> While the f and g are currently derived for MSE and logistic regression, it can be extended to models that can be locally linearized. We intend to provide more results in the future.
>
> **Q2: Presentation of the paper is incomplete, please define notations before using them even if they are conventional, especially for key results. E.g., what does the notation permute(1, L) and function H(\cdot) mean in Algorithm 2? What does -sensitive hashing function family mean in Theorem 2.**
>
> The  $permute(1, L)$ means the permutation list from $1$ to $L$. Function $H$ represents the  Simhash function. The $(S_0,cS_0,p_1,p_2)$ follows the definition of locality sensitive hash functions shown in Definition 1, appendix A.
>
> **Q3: The test error for slice test in the plot (h) of figure 4 is way higher than the training error, why?**
>
> We double-check the results. It may come from the limitation of linear models in regression.
>
> **Minor comments:**
> We thank the reviewer for carefully reading the paper and providing valuable suggestions. We will revise the paper by :(1) separating related work and preliminaries (2) fixing the reference and figures, (3) revise the statements

---

### Official Review · Reviewer_ycpC · 2021-07-14

**Rating:** 6
**Confidence:** 4

**Summary:**

The paper applies locality sensitive hashing to online teaching, where the machine uses a student’s information to retrieve a data sample from hash tables. The paper claims several contributions. (1) Propose a novel teaching framework, called Locality Sensitive Teaching (LST) for the retrieval. (2) LST has provable near-constant time complexity and is exponentially better than existing baseline (i.e. linear scan) (3) LST achieves 425.12x speedup over the linear scan baseline.

**Limitations And Societal Impact:**

The proposed method has potential contributions to energy reduction and privacy preserving. The authors claimed no negative impact to the society.

**Main Review:**

CT image, as in section 3.1 of the paper. Specifically, the paper formulates similar images retrieval regarding a query image as maximum inner product search of the query image and image database. With this formulation, the paper extends similarity retrieval to a practical domain.

Despite the above merits, the paper needs further improvements in the techniques and experiments to justify the claimed contributions.

1.	The paper claims LST is novel. But the paper does not explain the differences between LST and a related work ALSH. It seems that the paper simply applies ALSH asymmetric transformation to similar images retrieval. It will be good if the paper can explicitly point out the novel designs in LST that are not in ALSH.

2.	The paper claims near-constant complexity of LST as a key contribution in the Abstract. I can only find two appearances of words “constant” in the paper and cannot find the proof for “near-constant complexity”. If the paper means “sub-linear complexity”, than it is the contribution of ALSH and many pioneering works in maximum inner product search.

3.	The paper claims 425.12x speedup over baseline but only uses linear scan as baseline. Linear scan is a weak baseline in the field of maximum inner product search. The authors may compare LST with more state-of-the-art baselines such as ALSH.

[Comments regarding author response]
Thank you for the author response. Maybe I was biased towards the baselines so I requested more experiments evaluating existing algorithms for MIPS in the original review. I have increased my rating from 5 to 6 and I am happy to see this paper accepted.

**Time Spent Reviewing:**

3 hours

---

> ### Author Response · Authors · 2021-08-10
> **Thank you for the review, please see the following clarifications.**
>
> We thank the reviewer for the comments!  It seems the review is incomplete due to some glitch in the first sentence “CT image, as in section 3.1 of the paper.”. We hope it didn't miss something very important.
>
> We would like to stress that we use advances in LSH as a starting tool to obtain LST. To solve the problem, we have many contributions including novel transformation (which are significantly different from the ALSH methods known), making the math of LST amenable to an adaptive inner product sampling formalism, and experiments to validate the theory of LST.
>
> We provide justification of our paper regarding the three points in the review:
>
> 1. In this paper, we use novel asymmetric transformations that allow us to reformulate the greedy teaching algorithm into an inner product sampling problem.  The known ALSH transformations instead convert the maximum inner product search problem into a nearest neighbor search problem. Therefore, our asymmetric transformations and  ALSH transformations are different as they solve different problems.
>
> 2. In this work, we use LSH data structure for sampling. In ALSH for inner product search, we need to rerank the candidate vectors retrieved from hash buckets. However, in our work, we perform random sampling over the retrieved items. Theoretical results suggest that the sampling of retrieved items serves as a great kernel density estimator[1,3], local density estimator[2], and mutual information estimator[4].
>
> 3. The major contribution of our work is to reformulate greedy teaching algorithms as an inner product sampling problem and use LSH type data structure to solve it efficiently with theoretical guarantees.  We are open to other maximum inner product search (MIPS) data structures and hope our work would open the door for applying efficient MIPS data structures for machine teaching.  Currently, we have two major concerns: (1) MIPS data structures other than LSH have few theoretical guarantees on the search quality, (2) whether other MIPS data structures could be used to perform sampling remains an open question.
>
> We hope that the reviewer will reconsider the opinion that the contribution of our paper is different from ALSH.
>
> [1]Charikar, M., & Siminelakis, P. (2017, October). Hashing-based-estimators for kernel density in high dimensions. In 2017 IEEE 58th Annual Symposium on Foundations of Computer Science (FOCS) (pp. 1032-1043). IEEE.
>
> [2]Wu, X., Charikar, M., & Natchu, V. (2018, July). Local density estimation in high dimensions. In International Conference on Machine Learning (pp. 5296-5305). PMLR.
>
> [3]Coleman, B., & Shrivastava, A. (2020, April). Sub-linear race sketches for approximate kernel density estimation on streaming data. In Proceedings of The Web Conference 2020 (pp. 1739-1749).
>
> [4]Spring, R., & Shrivastava, A. (2020, July). Mutual Information Estimation using LSH Sampling. In IJCAI (pp. 2807-2815).

---

### Official Review · Reviewer_2Qsa · 2021-07-17

**Rating:** 8
**Confidence:** 4

**Summary:**

The paper proposes locality sensitive teaching (LST) which can be applied to IoT and online personalized education. In iterative machine teaching (IMT), the teacher selects example in each iteration for the student to learn, based on a greedy search algorithm which is expensive. The authors reformulate the optimization into a maximum inner product search problem, and use locality sensitive hashing (LSH) in the searching process to reduce the iteration-wise cost from $O(n)$ to $O(1)$. Theoretical analysis and practical hardware design are provided. Experiments validate the advantage of LST.

**Main Review:**

The paper is well organized and the writing is clear. The idea and derivation in each step is easy to follow.

Online machine teaching problem is important in practice and has many good applications. LSH is a classical method in information retrieval. Applying LSH to machine teaching leads to a substantial efficiency improvement, which is a solid contribution on this topic.

I like this paper since the formulation of the original optimization objective to MIPS is interesting and novel. Then the idea to use LSH is straightforward and appropriate. The method is described clearly, and the theory is adequate to justify the proposed LST. Also, it is good to discuss the hardware (GPU-CPU) design, which leads to further practical benefit and is helpful for the practitioners. The experiments are convincing, and the speedup is significant. Thus, I think this work can make a good contribution to NeurIPS.

Detailed comments and questions:

1) Typos on line 115 and 195.

2) In Algorithm 1, $\pi$ is not defined in the first loop; In Alg 2, $|B|$ is not used.

3) Is (2) and Theorem 2 strict for both regression and classification? Is it possible to extend Theorem 2 to more models?

**Time Spent Reviewing:**

2

---

> ### Author Response · Authors · 2021-08-10
> **Thank you for the comprehensive review, please see the following clarifications.**
>
> We appreciate your concise and precise summarization of our work!
>
> **Q1: Typos on lines 115 and 195:**
>
> Thanks for your help in improving our paper. We will revise the draft with a clearance of typos.
>
> **Q2: In Algorithm 1, $\pi$ is not defined in the first loop; In Alg 2,  $B$ is not used.**
>
> In Algorithm 1: $\pi$ is the permutation list from $1$ to $L$. In Algorithm 2, we use $B$ as the size of the bucket and perform a random sample over the bucket.
>
> **Q3: Extend Theorem 2 to more models:**
>
> We thank the reviewer for mentioning this great point.  We believe LST could be extended to models that can be locally linearized. We intend to present more results in future work.

---

### Official Review · Reviewer_QqeN · 2021-07-19

**Rating:** 6
**Confidence:** 3

**Summary:**

By reformulating Iterative Machine Teaching’s (IMT) optimization problem to an adaptive sampling problem, the papers proposes a Locality Sensitive Teaching (LST) algorithm. LST achieves exponential teachability and reduces the time complexity of each iteration from O(N), where N is the number of teaching examples, to few O(1) lookups in hash tables. They also attach the IoT perspective to the problem.

**Ethical Concerns:**

Paper plagiarism is 11% at Docoloc: https://www.docoloc.de/d88c4d5162dd4b21eeb6add446e2b5bbCEuFHG85rulAsphkx9/en/konto.hhtml?dogetresult=51. So it is OK - cleared as green signal.

**Limitations And Societal Impact:**

Not applicable, Skip.

**Main Review:**

Novelty:
The paper claims LST as the first algorithm that enables energy and time efficient machine teaching on IoT devices.

Strengths:
1. The paper focuses on 2 aspects - improving running metrics like speed, energy. And, overcoming greedy search of usual IMT systems.
2. The paper uses both theoretical and empirical approach as defense of the work. This seems to be concrete.
3. The paper's handling of concepts at mathematical rigor (backed by supplementary) is commendable.
4. Paper is well written.

Weakness:
1. Code is not available to check due to some permission issue.
2. The performance objective of the task - its accuracy - how that can be improved is not focused on. The trade-off between time/energy and accuracy needs a discussion.
3. The scalability section needs some validation - is this the correct approach to validate?
4. Adherence to practical usage (apart from Covid theme) should have increased the paper's value.
5. Mixing of IoT with the approach needs some more gelling from reader's continuity of flow perspective.

Relevance to NeurIPS:
More relevant to ICML/AAAI/KDD type conferences in contrast to Neural Information Processing. In fact IoT and EdTech conferences will also appreciate this work. However, for a change of inter-disciplinary flavor, bringing in the IoT perspective in ML is appreciated.

Effort: Significant effort has been done in thought process, however implementation wise would have like more rigor.

Suggestions:
1. Does the toy example in Fig.3 substantiate the approach's efficacy in terms of generalization?
2. Some discussion on bias / priors will help clarify the learning part.
3. How collision in the hash table can downgrade performance? Any heuristics to support?
4. Can adversarial examples affect the algorithm?
5. "D Motivating Example Settings" - this could have been expanded more to get an insight in the practical application.

**Time Spent Reviewing:**

3

---

> ### Author Response · Authors · 2021-08-10
> **Thank you for the comprehensive review, please see the following clarifications.**
>
> Clarification:
> 1. **Code availability:** The code is ready for sharing. We will release the codebase on github.
>
> 2. **Time/energy and accuracy trade-off:** We fix the target accuracy and perform an ablation study of speedup vs energy savings. Below is the snapshot at different levels. We will add this ablation study to the paper.
>
> Speedup (times),  Energy Saving(%)
>
> 140x speedup,  99.33% energy savings
>
> 130x speedup, 99.32%  energy savings
>
> 120x speedup, 99.20%  energy savings
>
> 110x speedup, 99.18%  energy savings
>
> 100x speedup, 99.17%  energy savings
>
> 3. **Scalability session:** We understand that the true measure of scalability is challenging. Therefore, in our experiments, we deployed it on a real IoT device, and further, we had multiple evaluation metrics which include both the energy measurement and speedup. Currently, TX2 is one of the popular devices for machine learning on IoT devices. We follow the popular hardware configuration of TX2 for IoT applications and provide measurement details in Appendix G. Therefore, we believe our experiment will directly translate into practice.
>
> 4. **Adherence to practical usage:** We will add the above discussion about practical scalability in the paper.
>
> 5. **Mixing of IoT with the approach:** We will revise the paper with more details on the IoT perspective of machine teaching in the lines mentioned above.
>
>
> Details on suggestions:
> 1. **Does the toy example in Fig.3 substantiate the approach's efficacy in terms of generalization?**
>
> In the example, we target minimizing the classification error in an example set of 4 points. We provide testing results by adding 10 test examples from the domain and evaluate the model. We observe that both AIPS and optimal solution achieves 90% accuracy, which is better than the 80% accuracy of GTA.  We will add these results and discussion in the paper.
>
> 2. **Some discussion on bias / priors will help clarify the learning part.**
>
> We understand this question as to the suggestion for discussing the bias and priors of optimal model $w^*$.  Given the $w^*$, we use LSH to perform importance sampling on the prior. Would you mind elaborating more on this question so that we could provide more detailed information?
>
> 3. **How collision in the hash table can downgrade performance? Any heuristics to support?**
>
> Since we use LSH, more collisions imply strong clustering, which makes the problem easier. We understand the question as the reviewer would like to know more about what happens when there are undesired collisions in the hash tables. The undesired collisions are standard in hashing literature. Their effect is a negligible error $\epsilon$ with mean zero, (and variance shrinking with the size of hash tables). These collisions don’t hurt the efficacy of hashing algorithms both in theory and practice. We can provide more discussions and references in the paper if needed.
>
> 4. **Can adversarial examples affect the algorithm?**
>
> Thanks for the great suggestion. This is an important future work to look into. Adversarial examples changed some of our understanding of machine learning. Since our procedure is randomized (Stochastic decisions) compared to the original IMT, we believe that our method is more robust to perturbations than IMT.  We plan to demonstrate how LST could handle adversarial examples in the future.
>
> 5. **"D Motivating Example Settings" - this could have been expanded more to get an insight in the practical application.**
>
> This is an informative feedback. Glad to know that it will help. We were not sure how much we can trade practical illustration vs technical details in the limited pages of the paper. We hope to collaborate with the health organizations so that we could use LST on IoT devices for training experts in identifying covid ct images.

---

### Decision · Program_Chairs · 2021-09-27

**Decision:**

Accept (Poster)

**Comment:**

The paper presents a faster algorithm for iterative machine teaching, which repeatedly traverses the training set to find samples for the learner. The improvement is obtained by replacing linear scan over the data set with a sampling approach based on locality-sensitive hashing. Empirical evaluation shows substantial speedups, up to 2-3 orders of magnitude, and similar energy savings.